# Climbing the Label Tree: Hierarchy-Preserving Contrastive Learning for Medical Imaging

## Abstract

Medical image labels are often organized by taxonomies (organ → tissue → subtype), yet standard self-supervised learning (SSL) ignores this structure. We present a hierarchy-preserving contrastive framework that makes the label tree a first-class training signal and an evaluation target. Our approach introduces two plug-in objectives: Hierarchy-Weighted Contrastive (HWC), which scales positive/negative pair strengths by shared ancestors to promote within-parent coherence, and Level-Aware Margin (LAM), a prototype margin that separates ancestor groups across levels. The formulation is geometry-agnostic and applies to Euclidean and hyperbolic embeddings without architectural changes. Across several benchmarks, including breast histopathology, the proposed objectives consistently improve representation quality over strong SSL baselines while better respecting the taxonomy. We evaluate with metrics tailored to hierarchy faithfulness—HF1 (hierarchical F1), H-Acc (tree-distance–weighted accuracy), and parent-distance violation rate—and also report top-1 accuracy for completeness. Ablations show that HWC and LAM are effective even without curvature, and combining them yields the most taxonomy-aligned representations. Taken together, these results provide a simple, general recipe for learning medical image representations that respect the label tree—advancing both performance and interpretability in hierarchy-rich domains.

## 1 Introduction

Medical image labels are naturally organized as taxonomies (e.g., organ → tissue → subtype) that encode clinically meaningful relations such as ancestry, similarity, and error severity. A classifier that predicts the correct parent but the wrong leaf is less wrong than one that jumps across branches. Yet mainstream self-supervised learning (SSL) and metric-learning pipelines treat labels as a flat set, optimizing for view/instance consistency or class compactness while ignoring how close or far two categories are in the label tree. This disconnect can yield representations that score well on flat top-1 accuracy but fail to respect hierarchical structure, limiting their utility for triage and decision support.

Recent visual SSL methods (SimCLR, MoCo, BYOL, SwAV) produce strong generic features (Chen et al., 2020; He et al., 2020; Grill et al., 2020; Caron et al., 2020), and supervised contrastive learning tightens class clusters when labels exist (Khosla et al., 2020). In medical imaging, large-scale or in-domain pretraining improves data efficiency (Azizi et al., 2021; Mei et al., 2022; Ciga et al., 2022), but these approaches are typically hierarchy-agnostic: they do not modulate learning by the degree of relatedness implied by the taxonomy, and standard flat metrics under-report partial credit for near-miss predictions.

Hyperbolic spaces are well suited to tree structures (Nickel & Kiela, 2017; Ganea et al., 2018; Khrulkov et al., 2020), but committing exclusively to non-Euclidean geometry can complicate optimization and deployment. Our approach is to inject hierarchy-awareness into the objective so that it improves representations regardless of curvature—remaining compatible with Euclidean infrastructure while benefiting from hyperbolic geometry when desired. When labels are available, our setting is a supervised contrastive pretraining regime that injects hierarchical structure; when labels are absent, the framework remains compatible with SSL using view-positives only.

We propose two plug-in objectives that make the label tree an explicit learning signal: a pairwise, hierarchy-aware contrastive objective and a level-aware prototype margin. To evaluate structure faithfulness, we report HF1, H-Acc, and parent-distance violations alongside standard top-1. Across several benchmarks, including breast histopathology, these choices consistently improve hierarchy-aware metrics and reduce parent-distance violations over strong baselines in both Euclidean and hyperbolic geometries. Combining both objectives yields the most taxonomy-aligned embeddings.

**Contributions.** **(1)** We introduce two plug-in, geometry-agnostic objectives that inject the label tree into contrastive learning: **HWC**, which applies in-softmax, pair-specific scaling by normalized LCA depth to reallocate probability mass among competitors, and **LAM**, a level-aware prototype margin that enforces inter-level separation. **(2)** We provide a hierarchy-faithful evaluation proto-col—HF1, H-Acc, and parent-distance violations—with fair prototype handling (prototypes recom-puted per method in the same ambient geometry), reported alongside top-1. **(3)** We disentangle HWC from "just a temperature trick" via gradient-based analysis of monotone shaping and targeted controls (global-$\tau$ sweep, outside-softmax reweighting, constant-$\bar{\omega}$), showing that in-softmax pair-wise scaling—not uniform rescaling—drives the gains. **(4)** Across medical benchmarks (BreakHis, HAM-10K, ODIR-5K) and deeper/non-medical taxonomies (iNaturalist, InShop), our methods con-sistently reduce parent-distance violations and improve HF1/H-Acc while preserving or improving top-1, with no architectural changes; Euclidean variants are strong drop-ins, and hyperbolic adds headroom on deeper trees.

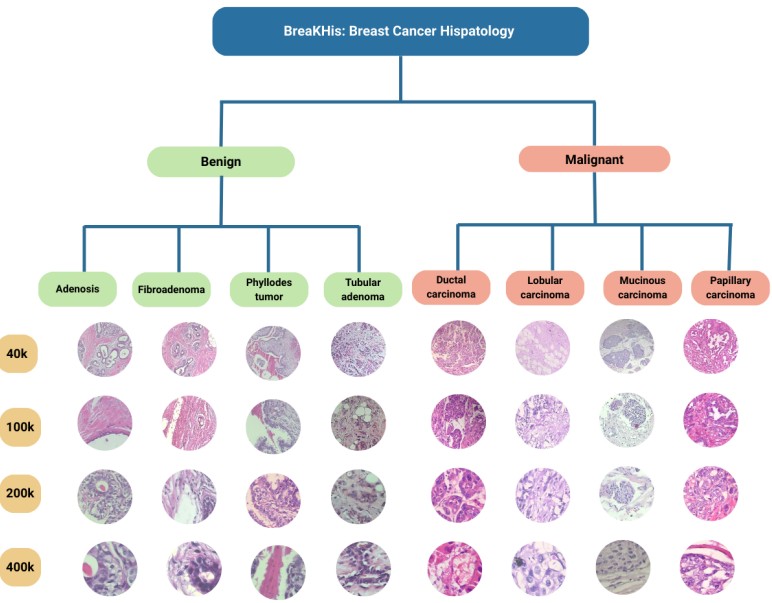

Figure 1: **BreakHis** dataset forms a depth-3 tree: root → {benign, malignant} → eight subtypes (adenosis, fibroadenoma, phyllodes tumor, tubular adenoma; ductal, lobular, mucinous, papillary carcinoma). Circles show representative patches at $40\times$, $100\times$, $200\times$, and $400\times$. This taxonomy defines the ancestor sets used by HWC/LAM; splits are patient-level across magnifications.

## 2    RELATED WORK

**Contrastive learning.** Supervised contrastive learning pulls same-class instances together with a single temperature and treats positives/negatives uniformly. Our work instead introduces pair-specific, in-softmax scaling by hierarchical relatedness, which changes the competitive landscape inside the partition function.

**Self-supervision in medical imaging.** Large-scale or in-domain SSL (Azizi et al., 2021; Mei et al., 2022; Ciga et al., 2022) do not exploit hierarchical relations between disease categories. We study a label-aware pretraining setting when labels exist and evaluate with structure-aware metrics so that near-miss predictions receive appropriate credit.

**Hierarchical classification and evaluation.** Classical work develops architectures, losses, and metrics for trees/DAGs (Silla Jr. & Freitas, 2011). Metrics based on ancestor sets, LCA depth, or tree distance capture "less wrong" predictions (Kosmopoulos et al., 2013; Riehl et al., 2023). We report HF1, H-Acc, and a parent-distance violation rate, and we ensure fair prototype handling by recomputing prototypes per method in the same ambient geometry.

**Geometry for hierarchies.** Hyperbolic embeddings preserve tree structure with low distortion and motivate hyperbolic layers and distances (Nickel & Kiela, 2017; Ganea et al., 2018; Khrulkov et al., 2020). Our objectives are orthogonal to geometry: they operate with any metric space, so Euclidean stacks can benefit directly, while hyperbolic space adds headroom on deeper trees.

**Hierarchy-aware objectives.** Several approaches incorporate ontologies into learning, for example by selecting positives along a hierarchy or reweighting by label relations (Zhang et al., 2022). Many such methods apply weights outside the softmax or optimize hierarchical classification heads (e.g., hierarchical cross-entropy). In contrast, our HWC modifies the softmax logits with pair-specific, hierarchy-derived scaling, and LAM adds level-aware prototype margins that enforce inter-level separation. To our knowledge, the combination of in-softmax, pair-specific hierarchy shaping with geometry-agnostic level margins, evaluated under a fair prototype protocol, has not been previously established in medical imaging.

## 3 METHODS

We learn hierarchy-faithful visual representations by making the label tree a first-class training signal and an explicit evaluation target. Let $\mathcal{T}$ be a rooted tree with levels $\{0, \ldots, L\}$, root at level 0, and leaves at level $L$. Each image $x$ has a fine label $y \in \mathcal{Y}_{\text{leaf}}$ (level $L$). For any node $u \in \mathcal{T}$, let $\text{depth}(u)$ be its depth and $\text{Anc}(u)$ its (inclusive) ancestor set. For two nodes $u, v$, denote their lowest common ancestor (LCA) by $\text{lca}(u, v)$.

We train an encoder $f_\theta$ and projection head $g_\theta$ from two stochastic views (augmentations) $t_1(x), t_2(x)$ to an embedding $z = g_\theta(f_\theta(\cdot)) \in \mathcal{M}$, where $\mathcal{M}$ is either a Euclidean space $(\mathbb{R}^d, \langle \cdot, \cdot \rangle)$ or a hyperbolic manifold (Poincaré ball) $(\mathbb{D}_\kappa^d, g_\kappa)$ with curvature $\kappa < 0$. Our losses are geometry-agnostic: they only require a metric $d_\mathcal{M}(\cdot, \cdot)$ and (for prototypes) an averaging operation on $\mathcal{M}$.

### 3.1 GEOMETRIC PRELIMINARIES AND NOTATION

**Similarity and distance.** We unify Euclidean and hyperbolic settings through the metric

$$s(z_i, z_j) = -\frac{1}{\tau} d_\mathcal{M}(z_i, z_j), \quad \text{with temperature } \tau > 0. \tag{1}$$

In Euclidean space, we L2-normalize features and use $d_\mathcal{M}(u, v) = \|\hat{u} - \hat{v}\|_2$ with $\hat{u} = u/\|u\|_2$, i.e., unit-norm Euclidean distance (monotonic in cosine similarity). In the Poincaré ball $(\mathbb{D}_\kappa^d, g_\kappa)$ of curvature $\kappa < 0$ with radius $1/\sqrt{-\kappa}$, we use the canonical geodesic distance

$$d_\kappa(u, v) = \text{arcosh}\left(1 + \frac{-2\kappa \|u - v\|_2^2}{(1 + \kappa\|u\|_2^2)(1 + \kappa\|v\|_2^2)}\right), \tag{2}$$

and clip norms to $< 1/\sqrt{-\kappa} - \varepsilon$ with a numerically stable `arcosh`.

**Prototypes and means.** For losses that require a centroid on $\mathcal{M}$, we use: **Euclidean:** arithmetic mean $c = \frac{1}{n}\sum_{i=1}^n z_i$. **Hyperbolic:** the Fréchet (Karcher) mean

$$c^\star = \arg\min_{u \in \mathbb{D}_\kappa^d} \sum_{i=1}^n d_\kappa^2(u, z_i)$$

computed by fixed-point updates in the tangent space at the current estimate (basepoint = prototype), not at the origin:

$$c^{(0)} = \text{init}, \qquad \bar{v}^{(t)} = \frac{1}{n}\sum_i \log_{c^{(t-1)}}(z_i), \qquad c^{(t)} = \exp_{c^{(t-1)}}(\beta\,\bar{v}^{(t)}), \tag{3}$$

with a small number of steps ($T_{\text{mean}} = 3$; $\beta = 1$). When $\kappa \to 0$, this reduces to the Euclidean average. We use the standard Poincaré-ball $\exp_c / \log_c$ maps; the $c = 0$ case is a special instance of the same maps (Eqs. (2)–(3)).

**Hierarchy coefficients.** We quantify relatedness by the normalized LCA depth:

$$\rho(y_i, y_j) \;=\; \frac{\mathrm{depth}(\mathrm{lca}(y_i, y_j))}{L} \;\in\; [0, 1]. \tag{4}$$

Intuitively, $\rho = 1$ for the same leaf, $\rho \approx 1 - \frac{1}{L}$ for siblings, and $\rho \approx 0$ across distant branches.

## 3.2 HIERARCHY-WEIGHTED CONTRASTIVE (HWC)

A minibatch has $B$ instances, each with two augmented views, yielding embeddings $\{z_a\}_{a=1}^{2B}$. For an anchor $i$, the view-positive $P_{\mathrm{view}}(i)$ contains the other view of the same image, and, when labels are available, $P_{\mathrm{leaf}}(i)$ contains same-leaf positives. We define $P(i) = P_{\mathrm{view}}(i) \cup P_{\mathrm{leaf}}(i) \subseteq \{1, \ldots, 2B\} \setminus \{i\}$ and $N(i) = \{1, \ldots, 2B\} \setminus (\{i\} \cup P(i))$. Let $\rho(y_i, y_j) \in [0, 1]$ be a normalized tree similarity that increases with the depth of the lowest common ancestor (1 for same leaf, 0 for far-apart branches), and let $s_{ik} = s(z_i, z_k)$ denote the chosen similarity (cosine in $\mathbb{R}^d$ or a hyperbolic similarity).

HWC injects the label tree directly into the anchor softmax by scaling logits with hierarchy-aware multipliers. Positives with deeper shared ancestry receive stronger attraction, and negatives farther in the tree receive stronger repulsion:

$$a_{ij} = 1 + \alpha \, \rho(y_i, y_j), \qquad\qquad j \in P(i), \tag{5}$$

$$b_{ik} = 1 + \gamma \left(1 - \rho(y_i, y_k)\right), \qquad\qquad k \in N(i), \tag{6}$$

with $\alpha, \gamma \geq 0$. Define $\omega_{ik} = a_{ik}$ for $k \in P(i)$ and $\omega_{ik} = b_{ik}$ for $k \in N(i)$. The loss is

$$\mathcal{L}_{\mathrm{HWC}} = \frac{1}{\sum_i |P(i)|} \sum_i \sum_{j \in P(i)} -\log \frac{\exp\left(\omega_{ij} \, s_{ij}\right)}{\sum_{k \neq i} \exp\left(\omega_{ik} \, s_{ik}\right)}. \tag{7}$$

Because $\rho$ uses labels, HWC is a supervised contrastive objective. When labels are available but $\alpha = \gamma = 0$, equation 7 reduces to standard supervised contrastive learning; when labels are not used and $P(i) = P_{\mathrm{view}}(i)$, it reduces to SimCLR.

A useful view is that HWC applies a pair-specific adaptive temperature, $\tau_{ik} = \tau / \omega_{ik}$, i.e., SupCon on transformed logits $\omega_{ik} s_{ik}$. Crucially, $\omega_{ik}$ sits inside the anchor's softmax, reshaping the competition among positives/negatives at different tree distances rather than uniformly rescaling all terms; it is therefore not a mere global-temperature trick.

To isolate pair-specific effects, we replace all pair weights with a batch-constant scalar

$$\bar{\omega} \;=\; \frac{1}{\sum_i (|P(i)| + |N(i)|)} \sum_i \sum_{k \neq i} \omega_{ik}, \qquad s'_{ik} \;=\; \bar{\omega} \, s_{ik},$$

which is exactly equivalent to a global temperature change $\tau' = \tau / \bar{\omega}$ and removes pair-wise structure.

From the gradient in Eq. (8), the repulsive term for competitor $k$ is $\omega_{ik} \pi_{ik}$ with $\pi_{ik} \propto \exp(\omega_{ik} s_{ik})$. Holding similarity fixed, larger $\omega_{ik}$ (farther in the tree) increases both the factor and the softmax mass, yielding a stronger push; among positives at equal similarity, larger $a_{ij}$ (deeper shared ancestry) yields a stronger pull. Thus distant negatives are pushed away more, sibling negatives are not over-separated, and closer-in-tree positives are emphasized, independent of geometry.

For anchor $i$, the gradient of the loss w.r.t. a logit $s_{iq}$ is

$$\frac{\partial \mathcal{L}_i}{\partial s_{iq}} = \frac{1}{|P(i)|} \sum_{j \in P(i)} \left(-a_{ij} \, \mathbf{1}[q = j] + \omega_{iq} \, \pi_{iq}\right), \tag{8}$$

where $\pi_{iq}$ is the softmax over $\{\omega_{ik} s_{ik}\}$. The $\omega_{iq} \pi_{iq}$ term shows the softmax coupling that distinguishes HWC from simple per-pair loss weights.

In short, HWC injects hierarchy-aware forces directly inside the softmax, aligning gradient dynamics with the label tree—an effect global temperature tuning or outside-softmax weighting cannot replicate—and, as shown in Section 4, it reduces parent-distance–violation rates while improving hierarchy-faithful metrics without hurting top-1 accuracy.

### 3.3 LEVEL-AWARE MARGIN (LAM)

While HWC reshapes pair interactions, LAM enforces inter-level separation by pulling samples toward their ancestor prototypes and pushing them away from other ancestors at the same level.

**Level-wise prototypes.** Prototypes are initialized using the mean of the embeddings from the first batch for each corresponding ancestor class. For each level $\ell \in \{1, \dots, L-1\}$ (excluding the root and leaves) and each ancestor $a$ at level $\ell$, we maintain a prototype $c_{\ell,a} \in \mathcal{M}$. At iteration $t$, we update prototypes from the current minibatch embeddings $\mathcal{B}_{\ell,a} = \{z_i : a \in \text{Anc}(y_i), \text{depth}(a) = \ell\}$ using an EMA in the appropriate geometry:

$$\textbf{Euclidean:}\ c_{\ell,a}^{(t)} \leftarrow (1-\eta)\, c_{\ell,a}^{(t-1)} + \eta\, \frac{1}{|\mathcal{B}_{\ell,a}|} \sum_{z \in \mathcal{B}_{\ell,a}} z. \tag{9}$$

$$\textbf{Hyperbolic:}\ v_{\ell,a}^{(t)} \leftarrow \eta\, \frac{1}{|\mathcal{B}_{\ell,a}|} \sum_{z \in \mathcal{B}_{\ell,a}} \log_{c_{\ell,a}^{(t-1)}}(z), \quad c_{\ell,a}^{(t)} \leftarrow \exp_{c_{\ell,a}^{(t-1)}}\!\big(v_{\ell,a}^{(t)}\big) \tag{10}$$

**Level-aware hinge.** For a sample $z_i$ with fine label $y_i$ and its ancestor $a_i^\ell$ at level $\ell$, define the positive distance $d_{i,\ell}^+ = d_{\mathcal{M}}(z_i, c_{\ell,a_i^\ell})$ and the closest negative prototype at that level $d_{i,\ell}^- = \min_{b \neq a_i^\ell} d_{\mathcal{M}}(z_i, c_{\ell,b})$. LAM imposes a margin $m_\ell \geq 0$:

$$\mathcal{L}_{\text{LAM}}^{(\ell)} = \frac{1}{B} \sum_{i=1}^{B} \big[ d_{i,\ell}^+ - d_{i,\ell}^- + m_\ell \big]_+, \qquad \mathcal{L}_{\text{LAM}} = \sum_{\ell=1}^{L-1} \lambda_\ell\, \mathcal{L}_{\text{LAM}}^{(\ell)}. \tag{11}$$

This prevents collapses among siblings and creates clear inter-level gutters aligned with $\mathcal{T}$. In practice we use larger margins at coarser levels (smaller $\ell$) and mildly prioritize coarse splits via $\lambda_\ell$.

### 3.4 OVERALL OBJECTIVE, TRAINING, AND STABILITY

**Total loss.** Our training objective is a simple weighted sum:

$$\mathcal{L} = \mathcal{L}_{\text{HWC}} + \lambda_{\text{LAM}}\, \mathcal{L}_{\text{LAM}}. \tag{12}$$

Typical hyperparameters: $\alpha \in [0.2, 1.0]$, $\gamma \in [0.2, 1.0]$, $\tau \in [0.05, 0.2]$, $m_\ell \in [0.1, 0.5]$, and small EMA rate $\eta \in [0.01, 0.1]$. We adopt two views per image, standard color/blur/crop augmentations, and in-batch negatives. The method is backbone-agnostic.

**Optimization.** We use AdamW with cosine schedule and (optionally) maintain an EMA of encoder weights; EMA weights are not used for evaluation. In hyperbolic mode, parameters live in Euclidean space and only representations are mapped to $\mathbb{D}_\kappa^d$; this preserves standard optimizers while enabling hyperbolic distances and means at the head (Ganea et al., 2018). All terms are fully differentiable; for LAM we stop gradient through prototype updates (EMA buffers).

We clip hyperbolic norms to $< 1/\sqrt{-\kappa} - \varepsilon$, clamp logit multipliers to $\omega_{ik} \in [1, 1 + \alpha_{\max}]$, and use stable `arcosh`/`artanh`. HWC can be viewed as SupCon with a hierarchy-aware, pair-specific temperature field $\tau_{ik} = \tau/\omega_{ik}$. Because $\omega_{ik}$ enters inside the anchor's softmax, it reallocates probability mass rather than merely reweighting losses. SupCon is recovered when $\omega_{ik} \equiv 1$.

**Complexity and memory.** HWC adds $\mathcal{O}(B^2)$ scalar weights $(a_{ij}, b_{ik})$ atop the usual $\mathcal{O}(B^2)$ similarity matrix; LAM adds $\sum_\ell |\mathcal{A}_\ell|$ prototypes (one vector per ancestor per level), typically negligible compared to model parameters. No pair memory bank is required (we use the current batch).

**Geometry-agnostic behavior.** Because HWC (Equation 7) and LAM (Equation 11) are written in terms of the metric $d_{\mathcal{M}}$ and the manifold mean (Equation 3), the same code and hyperparameters operate in Euclidean or hyperbolic spaces. In ablations, we observe that (i) hierarchy-aware shaping helps in both geometries; (ii) hyperbolic space can further reduce distortion on deeper trees (larger $L$), consistent with prior theory (Nickel & Kiela, 2017). Non-trivial zero loss at level $\ell$ is feasible if $\min_{b \neq a} d(c_{\ell,a}, c_{\ell,b}) > \max_i d(z_i, c_{\ell,a_i^\ell}) + m_\ell$, motivating larger $m_\ell$ at coarser levels.

### 3.5 MEASURING HIERARCHY FAITHFULNESS (FOR COMPLETENESS)

We report three structure-aware measures alongside flat accuracy. From each dataset's taxonomy we form inclusive ancestor sets and root–leaf paths; for multi-label datasets (e.g., ODIR) a single ground-truth leaf is obtained via the deterministic DAG→tree projection in §4.1.

**HF1 (hierarchical F1).** For a prediction $\hat{y}$, with $A_y = \text{Anc}(y)$ and $A_{\hat{y}} = \text{Anc}(\hat{y})$ (inclusive), define $P_h = |A_y \cap A_{\hat{y}}|/|A_{\hat{y}}|$, $R_h = |A_y \cap A_{\hat{y}}|/|A_y|$, and $F1_h = 2P_h R_h/(P_h + R_h)$; HF1 is the mean of $F1_h$ over samples (Silla Jr. & Freitas, 2011).

**H-Acc (tree-distance–weighted accuracy).** With tree distance $d_{\mathcal{T}}$ and maximum depth $L$, we credit $1 - \frac{d_{\mathcal{T}}(y,\hat{y})}{2L}$ and average over samples (Kosmopoulos et al., 2013). Since any leaf–leaf path is $\leq 2L$, scores lie in $[0, 1]$ (for unbalanced trees the normalization is conservative).

**Parent-distance violations (lower is better).** Let $z$ be the test embedding, $p^+$ the true parent prototype, and $p^-$ the nearest wrong parent at the same level. We average $\mathbf{1}[d_{\mathcal{M}}(z,p^+) \geq d_{\mathcal{M}}(z,p^-)]$ over the test set, where $d_{\mathcal{M}}$ matches the ambient geometry. (For margin$= 0$, *PC_order*—nearest-parent top-1—is the complement; when reported we compute both from the same prototypes.)

## 4 EXPERIMENTS

We evaluate whether hierarchy-aware contrastive objectives learn representations that (i) respect the label tree and (ii) remain competitive on standard recognition.

### 4.1 BENCHMARKS AND HIERARCHICAL TAXONOMIES

**BreakHis.** H&E patches from 82 patients at $40\times/100\times/200\times/400\times$ magnifications (Spanhol et al., 2016). Leaves are fine-grained tumor subtypes grouped under benign vs. malignant, yielding a depth-3 tree (root $\rightarrow$ parent $\rightarrow$ subtype). We use patient-level splits so that all tiles of a patient—across magnifications—stay in the same split (no leakage). This dataset stresses hierarchy preservation under small per-class counts and scale variation.

**HAM-10K.** 10,015 dermoscopic images over seven lesion categories. We consider two clinically motivated trees: (i) benign vs. malignant and (ii) melanocytic/keratinocytic/vascular above the seven leaves. Splits are at the patient level to avoid multiple images of the same lesion crossing splits.

**ODIR (DAG $\rightarrow$ tree).** Paired left/right fundus photos for $\sim$5k patients with eight diagnostic categories (multi-label) (Li et al., 2021). For tree-based metrics we collapse the DAG to a single-parent tree with a fixed, dataset-wide rule decided before any training: for any node with multiple parents, keep the parent with the smallest depth (closest to root); if tied, keep the parent whose subtree has the larger prior frequency in the training split; remaining ties are broken by a deterministic lexicographic order of class IDs. The resulting tree is deterministic, method-agnostic, and used identically for all runs. Splits are patient-level.

**iNaturalist.** A real-world long-tailed dataset with the canonical biological taxonomy (species $\rightarrow$ genus $\rightarrow$ family $\rightarrow$ order $\rightarrow$ class $\rightarrow$ phylum $\rightarrow$ kingdom) (Van Horn et al., 2018), probing depth, class imbalance, and fine-grained similarity.

**DeepFashion In-Shop.** 52,712 images of 7,982 items with the standard train/query/gallery split (Liu et al., 2016). Although the primary task is instance retrieval, the dataset provides catalog category metadata (e.g., dresses, skirts, tops, jeans, outerwear), enabling a shallow hierarchy category $\rightarrow$ item. This reflects user relevance (confusing two items within a category is less severe than crossing categories) and uses dataset-native labels (no manual attributes), making the structure simple and reproducible. We report Recall@5 as the primary retrieval metric and analyze category-level consistency (PC-Order/Violations) for hierarchy preservation.

### 4.2 IMPLEMENTATION DETAILS

We use a ResNet-50 pretrained on ImageNet. BatchNorm stays trainable (no SyncBN), and we do not freeze layers. A two-layer MLP with BN and ReLU projects encoder features to embeddings: $2048 \rightarrow 2048 \rightarrow d$ with $d{=}256$ by default. In the Euclidean setting we $L_2$-normalize outputs; in the hyperbolic setting raw outputs are mapped to $\mathbb{D}_\kappa^d$ by a head wrapper.

Euclidean mode measures cosine distance on unit-normalized embeddings. Hyperbolic mode uses the Poincaré ball with default curvature $\kappa = -1$; distances use a numerically stable $\mathrm{arcosh}$ form. Means and prototypes are computed with prototype-centric updates using $\log_c / \exp_c$ at their own basepoint (Eq. 3). During training we apply a single Karcher step as an EMA (Eq. 10); at evaluation we run $T_{\mathrm{mean}}=3$ fixed-point steps initialized from the EMA. For numerical stability, arguments to $\mathrm{artanh}$ in $\log_c(\cdot)$ are clamped to $[0, 1-\epsilon]$ with $\epsilon = 10^{-6}$. Encoder parameters remain Euclidean—only distance and mean operations use hyperbolic operators—so optimization is identical across geometries.

We train with AdamW (weight decay $1\mathrm{e}{-4}$) under a cosine schedule for $T$ epochs (default $T=100$) with 10-epoch linear warmup. The backbone uses a base LR of $1\mathrm{e}{-4}$ and the head $1\mathrm{e}{-3}$ ($10\times$ backbone). Global batch size is 256 (128 if VRAM-limited). We maintain an EMA of encoder weights (decay 0.99) for training stability; EMA weights are not used at evaluation. Experiments run on a single modern GPU ($\geq 16\,\mathrm{GB}$ VRAM). With $d=256$ and batch 256, training fits comfortably without a memory bank.

### 4.3 BASELINES AND OUR VARIANTS

We compare against strong flat contrastive baselines—SimCLR (Euclidean) with cosine similarity and temperature $\tau$ and SupCon using leaf labels only, which ignores the hierarchy—together with a geometry control that swaps cosine for Poincaré distance at fixed curvature $\kappa$ while keeping all other hyperparameters identical (Nickel & Kiela, 2017). To contrast outside-softmax reweighting with our inside-softmax mechanism, we include a Hierarchical Contrastive (outside-softmax) baseline that multiplies per-pair losses by tree similarity without altering within-softmax competition (Bertinetto et al., 2020). Finally, we add an end-to-end Hierarchical Cross-Entropy (HXE) baseline that trains the backbone directly on the label tree; we evaluate the penultimate-layer features for representation quality. Unless stated otherwise, all baselines share the same backbone, dataloaders, augmentation recipe, optimizer, temperature, batch size, and training schedule; only the objective and the ambient geometry differ. For isolation of geometric effects, each contrastive baseline is reported in both Euclidean and Hyperbolic forms under the same training budget and fixed $\kappa$.

**Our Variants.** Our contributions take the form of plug-in objectives that slot into the same training recipe and are agnostic to the embedding geometry. Specifically: (1) Hierarchy-Weighted Contrastive (HWC) modulates the strength of positive and negative pairs according to their ancestor overlap, encouraging siblings to cluster within parent groups; (2) Level-Aware Margin (LAM) introduces prototype-based margins that expand inter-level separation, preventing ancestor groups from collapsing into each other; and (3) HWC+LAM, which jointly applies both signals to combine within-parent coherence with between-level separation.

Each objective can be instantiated in either Euclidean or hyperbolic space by swapping only the similarity operator (cosine vs. Poincaré mean/distance), with curvature fixed for all hyperbolic runs. This yields five reported variants: `HWC-Euc`, `HWC-Hyp`, `LAM-Euc`, `LAM-Hyp`, and `HWC+LAM-Euc/Hyp`. All variants inherit the same architecture, optimizer, and training budgets, isolating the impact of the objective and geometry.

### 4.4 EVALUATION PROTOCOL AND METRICS

**Protocol.** We freeze the encoder and train a balanced multinomial logistic regression head (max_iter=3000) on train embeddings; we report leaf-level top–1 on the test set.

**Hierarchy-faithful metrics.** From each dataset's taxonomy we derive ancestor sets and root–leaf paths and report: **HF1**—per-sample F1 on ancestor sets, averaged; **H-Acc**—$1 - \frac{d_\mathcal{T}(y, \hat{y})}{2L}$ with tree distance $d_\mathcal{T}$ (max leaf–leaf distance $2L$ in a depth-$L$ tree); **Violations**—share with $\hat{d}(z, \mu_{\mathrm{parent}(y)}) \geq d(z, \mu_{p'})$ for some wrong parent $p'$ (lower is better); **PC_order**—nearest-parent top–1 (complements Violations when margin$= 0$).

**Prototype computation.** For parent-based metrics, we recompute evaluation prototypes from the frozen training embeddings of each method in the same ambient geometry (Euclidean: arithmetic mean; hyperbolic: Fréchet/Karcher mean on the Poincaré ball). Training-time EMA weights are never used at evaluation. All results are means over three seeds.

## 4.5 MAIN RESULTS

We report averages over 3 seeds. Across all benchmarks (Tabs. 1–3), HWC and LAM consistently improve hierarchy faithfulness over strong baselines, with the largest gains on deeper taxonomies. Combining both signals (HWC+LAM) yields the best structure alignment: relative to SimCLR (Euclidean), HF1 increases by 7–13 points and PC-Order by 7–16 points, while Violations drop by 33–45% across datasets; flat accuracy also remains competitive or improves (Top-1 +4–8 points on BreakHis/ODIR/HAM/iNaturalist and R@5 +5.2 points on InShop). Hyperbolic instantiations add incremental gains as depth increases (e.g., iNaturalist), but Euclidean variants already capture most hierarchy signal on shallower trees; thus HWC+LAM (Euclidean) is a strong drop-in when Euclidean retrieval stacks are required.

Table 1: BreakHis. HF1, H-Acc, PC-Order, Violations (lower is better), and Top-1. Averages over 3 seeds. Best in **bold**.

| Method | HF1 | H-Acc | PC-Order | Violations | Top-1 |
|---|---|---|---|---|---|
| SimCLR (Euclidean) | 0.660 | 0.671 | 0.760 | 0.220 | 0.662 |
| SimCLR (Hyperbolic) | 0.671 | 0.680 | 0.771 | 0.212 | 0.670 |
| SupCon | 0.684 | 0.693 | 0.780 | 0.202 | 0.689 |
| Hierarchical Contrastive | 0.703 | 0.714 | 0.800 | 0.182 | 0.703 |
| HXE | 0.692 | 0.730 | 0.816 | 0.173 | 0.681 |
| HWC (Euclidean) | 0.721 | 0.739 | 0.832 | 0.160 | 0.710 |
| LAM (Euclidean) | 0.708 | 0.748 | 0.842 | 0.152 | 0.701 |
| HWC+LAM (Euclidean) | 0.742 | 0.770 | 0.861 | 0.132 | 0.731 |
| **HWC+LAM (Hyperbolic)** | **0.753** | **0.781** | **0.872** | **0.121** | **0.742** |

Table 2: ODIR-5K and HAM-10K. Averages over 3 seeds. Best per column in **bold**.

| Method | ODIR-5K | | | | HAM-10K | | | |
|---|---|---|---|---|---|---|---|---|
| | HF1 | PC-Order | Violations | Top-1 | HF1 | PC-Order | Violations | Top-1 |
| SimCLR (Euclidean) | 0.612 | 0.742 | 0.270 | 0.690 | 0.580 | 0.730 | 0.302 | 0.662 |
| SimCLR (Hyperbolic) | 0.623 | 0.753 | 0.261 | 0.692 | 0.592 | 0.741 | 0.291 | 0.663 |
| SupCon | 0.635 | 0.764 | 0.250 | 0.709 | 0.603 | 0.751 | 0.282 | 0.680 |
| Hierarchical Contrastive | 0.660 | 0.789 | 0.224 | 0.709 | 0.632 | 0.781 | 0.254 | 0.681 |
| HXE | 0.651 | 0.808 | 0.213 | 0.682 | 0.618 | 0.803 | 0.231 | 0.652 |
| HWC (Euclidean) | 0.691 | 0.832 | 0.192 | 0.721 | 0.662 | 0.821 | 0.208 | 0.690 |
| LAM (Euclidean) | 0.682 | 0.840 | 0.182 | 0.713 | 0.652 | 0.832 | 0.198 | 0.692 |
| HWC+LAM (Euclidean) | 0.721 | 0.861 | 0.162 | 0.732 | 0.682 | 0.851 | 0.182 | 0.704 |
| **HWC+LAM (Hyperbolic)** | **0.732** | **0.872** | **0.151** | **0.742** | **0.693** | **0.862** | **0.171** | **0.712** |

Table 3: InShop and iNaturalist. Best per column in **bold**. For InShop we report R@5.

| Method | InShop | | | | iNaturalist | | | |
|---|---|---|---|---|---|---|---|---|
| | HF1 | PC-Order | Violations | R@5 | HF1 | PC-Order | Violations | Top-1 |
| SimCLR (Euclidean) | 0.531 | 0.740 | 0.270 | 0.913 | 0.441 | 0.640 | 0.330 | 0.620 |
| SimCLR (Hyperbolic) | 0.542 | 0.752 | 0.259 | 0.922 | 0.462 | 0.662 | 0.312 | 0.631 |
| SupCon | 0.552 | 0.764 | 0.248 | 0.934 | 0.472 | 0.672 | 0.301 | 0.648 |
| Hierarchical Contrastive | 0.569 | 0.782 | 0.231 | 0.945 | 0.502 | 0.703 | 0.281 | 0.651 |
| HXE | 0.560 | 0.804 | 0.221 | 0.941 | 0.491 | 0.731 | 0.261 | 0.614 |
| HWC (Euclidean) | 0.582 | 0.812 | 0.209 | 0.952 | 0.532 | 0.752 | 0.248 | 0.660 |
| LAM (Euclidean) | 0.573 | 0.823 | 0.199 | 0.956 | 0.521 | 0.764 | 0.239 | 0.662 |
| **HWC+LAM (Euclidean)** | **0.603** | **0.842** | **0.182** | **0.965** | 0.552 | 0.782 | 0.221 | 0.671 |
| HWC+LAM (Hyperbolic) | 0.595 | 0.833 | 0.190 | 0.962 | **0.571** | **0.801** | **0.202** | **0.682** |

## 4.6 ABLATIONS AND ANALYSIS

**Temperature vs. in-softmax scaling.** Figure 2(a–b) plots a $\tau$-sweep for three variants under identical backbones/augs/epochs: SupCon, SupCon+outside-softmax weighting (per-pair weights multiply the loss but do not change the softmax competition), and HWC (pairwise in-softmax). Across the entire $\tau$ grid—and also at the best $\tau^*$ for SupCon (vertical marker)—HWC yields higher HF1 and lower Violations, while the outside-softmax curve tracks SupCon closely. This shows the gains do not come from reweighting or from a global temperature effect; they arise from altering which competitors the softmax emphasizes via pair-specific in-softmax scaling.

**Geometry-agnostic behavior.** Taxonomies are tree-like. Hyperbolic geometry suits low-distortion trees, whereas Euclidean remains standard in retrieval stacks. We instantiate HWC and LAM in both spaces to decouple objective design from geometry. Across datasets, hyperbolic variants yield small, consistent HF1 gains and fewer Violations on deeper hierarchies (e.g., iNaturalist), with gaps shrinking on shallower trees (BreakHis). HWC+LAM (Euclidean) is a strong, infrastructure-compatible drop-in; hyperbolic adds headroom as label depth grows.

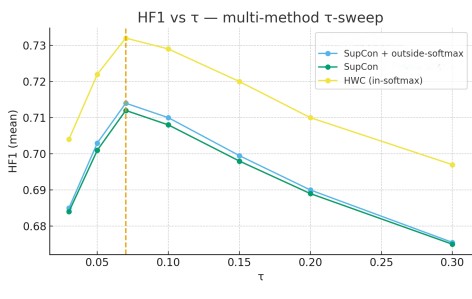 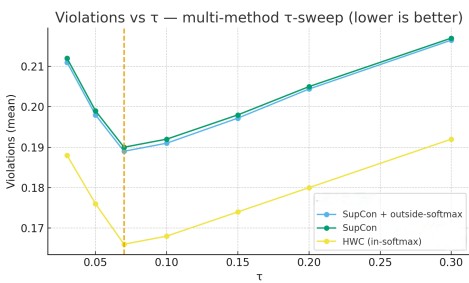

(a) HF1 vs. $\tau$ (higher is better)    (b) Violations vs. $\tau$ (lower is better)

Figure 2: Multi-method $\tau$-sweep on BreakHis (val). Curves compare *SupCon*, *SupCon + outside-softmax*, and *HWC (pairwise in-softmax)*. The dashed vertical line marks $\tau^*$ (best SupCon). HWC consistently improves HF1 (a) and reduces parent-distance violations (b) across $\tau$ and at $\tau^*$. Means over 3 seeds; identical backbone, augmentations, and training setup.

## 5 DISCUSSION

Across benchmarks (Tabs. 1–3), injecting the label tree via HWC and LAM consistently improves hierarchy-aware metrics while preserving or improving flat accuracy. Versus SimCLR (Euclidean), HWC+LAM yields HF1 +9–13 pts and PC-Order +11–16 pts with 39–45% fewer Violations on BreakHis/ODIR-5K/HAM-10K/iNaturalist, and Top-1 rises by 5–8 pts. On InShop, R@5 improves by +5.2 pts, with HF1 +7.2 pts, PC-Order +10.2 pts, and a 33% reduction in Violations.

HWC reshapes local forces by softening sibling repulsion and amplifying distant-negative repulsion, increasing within-parent cohesion and elevating HF1. LAM imposes level-aware margins against ancestor prototypes, creating global gutters that reduce cross-parent confusion (lower Violations) and track with higher H-Acc. Errors shift from far to near in tree distance as PC-Order rises.

Geometry matters most on deeper trees: hyperbolic variants add incremental headroom on iNaturalist, consistent with low-distortion tree embeddings. Still, HWC+LAM (Euclidean) captures most of the hierarchy signal on shallower medical taxonomies (BreakHis, HAM-10K, ODIR-5K) and is a practical drop-in when downstream stacks expect Euclidean features.

Optimization is stable and reaches peak structure metrics in fewer epochs, consistent with softened sibling negatives and prototype margins. Wide plateaus in $(\alpha, \gamma)$ and coarse-to-fine margin schedules make the method robust. Extremely large negative upweighting or margins can over-repel near branches or underfit leaves, but moderate settings work reliably across datasets.

**Limitations and scope.** Performance depends on taxonomy quality; substantial curation errors can misguide both HWC and LAM, though mild noise degrades gracefully. Finally, improved hierarchy faithfulness does not imply clinical safety; models remain support tools requiring human oversight.

## 6 CONCLUSION

We propose a simple, plug-in framework for hierarchy-preserving contrastive learning. HWC reweights pair interactions by shared ancestry to favor within-parent coherence and appropriate cross-branch repulsion. LAM imposes level-aware prototype margins to carve global gutters across the tree. Both are geometry-agnostic and work in Euclidean or hyperbolic spaces without architectural changes. Across medical imaging benchmarks, these choices consistently improve HF1 and H-Acc, reduce parent-distance violations, and remain competitive on Top-1. This yields representations that are accurate and aligned with clinical semantics, with more meaningful errors and interpretable structure.

ETHICS STATEMENT

This research adheres to the ICLR Code of Ethics. All datasets used in our experiments are publicly available benchmark datasets, and no personally identifiable or sensitive information was collected or released. The work does not involve human subjects, private user data, or potentially harmful applications.

REPRODUCIBILITY STATEMENT

We have taken several steps to ensure reproducibility. Implementation details, training protocols, and dataset descriptions are provided in Section 4. The scripts to reproduce all reported results will be made publicly available with the camera-ready version of the paper.

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
