# OpenReview forum: "Climbing the label tree: Hierarchy-preserving contrastive learning for medical imaging"
_ICLR.cc/2026/Conference — ICLR 2026 Conference Withdrawn Submission_

### Official Review · Reviewer_2v3S · 2025-10-29

**Soundness:** 3
**Presentation:** 2
**Contribution:** 2
**Rating:** 4
**Confidence:** 5

**Summary:**

This paper addresses the overlooked hierarchical structure of medical image labels (organ → tissue → subtype) often ignored in conventional SSL. The authors propose a hierarchy-preserving contrastive framework that leverages the label tree as both a training signal and an evaluation target. Experiments on five benchmarks demonstrate the effectiveness of the proposed method.

**Strengths:**

The paper is clearly written, with a precise problem formulation and compelling motivation. The proposed framework is conceptually sound and effectively highlights the limitations of existing SSL methods in medical images.

**Weaknesses:**

1. While the idea of hierarchical contrastive learning is intuitive, the method depends on datasets annotated at multiple granularity levels (e.g., both global categories and fine-grained subtypes), which restricts its general applicability.
2. The use of multiple level-wise prototypes introduces additional computational overhead—yet the paper lacks a detailed analysis of training efficiency.
3. There is no systematic discussion on how key hyperparameters (e.g., those mentioned around L250) are chosen or tuned.

**Questions:**

N/A

---

### Official Review · Reviewer_Asyc · 2025-10-30

**Soundness:** 3
**Presentation:** 2
**Contribution:** 3
**Rating:** 6
**Confidence:** 5

**Summary:**

The paper adds hierarchy awareness to contrastive learning via two plug-in losses: (1) Hierarchy-Weighted Contrastive (HWC), which scales each pair inside the softmax by how much ancestry two labels share, and (2) Level-Aware Margin (LAM), which pulls samples toward their ancestor prototypes and enforces level-wise margins. Both work in Euclidean or hyperbolic space without changing the backbone. On BreakHis, HAM-10K, ODIR-5K, iNaturalist, and In-Shop, the method improves hierarchical metrics (HF1, H-Acc, parent-distance violations) and usually keeps or improves flat accuracy. Ablations argue that gains come from the “inside-softmax” shaping rather than temperature tuning.

**Strengths:**

S1. Problem formulation - Flat objectives ignore how “near” or “far” two labels are in a label tree. The paper makes the tree both a training signal and an evaluation target, and uses patient-level splits on medical sets (e.g., BreakHis) to avoid leakage.

S2. Drop-in objectives - HWC and LAM slot into standard contrastive pipelines; swapping Euclidean vs. hyperbolic only changes the metric and mean operator, not the architecture or optimizer. This makes proposed method adaptable easily and can be helpful to community.

S3. The paper reports hierarchy-faithful metrics with fair prototype handling (recomputed per method in the same geometry) alongside top-1 / R@5, which improves interpretability. Across datasets, HWC+LAM improves HF1 and H-Acc and cuts parent-distance violations; Euclidean already works well on shallow trees, hyperbolic adds headroom as depth grows.

S4. Though novelty is limited but utility of method makes this work significant to the society and provide geometrical perspective.

**Weaknesses:**

W1. Marginal novelty of incremental nature -  Pair-wise hierarchy shaping and prototype margins are known ideas; the main step is to put hierarchy scaling inside the softmax and combine it with level margins. Positioning against prior hierarchical contrastive/margin literature could be useful in my understanding.

W2. Limited shift analysis - Medical data shift by site, stain, and magnification. The paper does not test cross-magnification or stain/center robustness, even though BreakHis spans 40×–400× and the splits are patient-level across magnifications. Results requires more investigation across datasets to support the method performance. Some of the works in past has focused on it such as Magnification prior (2023) focusing on cross resolution validations. It can strengthen the claim.
[1] Chhipa, P. C., Upadhyay, R., Pihlgren, G. G., Saini, R., Uchida, S., & Liwicki, M. (2023). Magnification prior: a self-supervised method for learning representations on breast cancer histopathological images. In Proceedings of the IEEE/CVF winter conference on applications of computer vision (pp. 2717-2727).

W3. Curvature choice is fixed - Hyperbolic runs use a fixed curvature; no study of learned curvature or sensitivity is provided. This weakens the “geometry-agnostic” claim on deeper trees. A detailed analysis will be helpful.

W4. LAM uses EMA prototypes without gradient flow. Stability vs. EMA rate, margins per level, and small-class effects are not deeply explored. A goal oriented discussion shall allow research community to understand the trade-offs.

W5. ODIR DAG to tree projection is deterministic but simplistic; the impact of this choice on metrics and training is not described.

**Questions:**

Along with weaknesses above, following are additional questions -

Q1. How do results change under cross-magnification splits on BreakHis (e.g., train on 40×/100×, test on 200×/400×)? Please report HF1/H-Acc and violations vs. top-1.

Q2. Have you tried learned curvature or per-layer curvatures in the hyperbolic variant, and how sensitive are results to curvature and embedding dimension?

Q3. For LAM, what EMA rates and margin schedules work best for small classes? Any failure cases with sibling collapse or oscillation?

I am willing to adjust my score based on potential addressing of mentioned issues.

---

### Official Review · Reviewer_VZet · 2025-10-31

**Soundness:** 3
**Presentation:** 3
**Contribution:** 2
**Rating:** 4
**Confidence:** 4

**Summary:**

This paper proposes a hierarchy-aware contrastive representation learning framework. The paper introduces a Hierarchy-Weighted Contrastive (HWC) loss that scales pairwise logits inside the softmax according to the LCA depth and a Level-Aware Margin (LAM) loss that defines ancestor prototypes and imposes level-dependent margins. The approach is geometry-agnostic, compatible with both Euclidean and hyperbolic embedding spaces, and evaluated across several medical and natural image benchmarks. Experimental results demonstrate improved hierarchical metrics.

**Strengths:**

The paper convincingly motivated by hierarchical medical labels.

The proposed method is tested on multiple datasets and observe consistent performance gains.

**Weaknesses:**

There have been other related work that closely related to hierarchy-aware contrastive learning [1,2] that are not discussed in the manuscript, and some of the methods look similar to the proposed method. The authors might need to clarify how the proposed method differs from existing methods and compare with them quantitatively in the experiments if possible.

Although this manuscript is motivated by medical imaging, the proposed method itself does not look medical-specific. The loss formulation is general and applies to any hierarchical taxonomy. In fact, the experiments include non-medical benchmarks.  While this generality is positive, the manuscript could better clarify whether the method leverages any medical-domain structure or if the contribution is purely in general hierarchical contrastive learning.

This work specifically aims to learn hierarchy-preserving representations, yet the paper does not provide any visualization of the learned embedding space. For a contrastive representation learning method whose core contribution is structuring embeddings according to a label tree, qualitative visualization would be particularly valuable. Embedding plots reveal whether samples belonging to the same parent node cluster together and how higher-level taxonomy structure is reflected in the embedding geometry.

References:

[1] Yu, Simon Chi Lok, Jie He, Victor Gutierrez Basulto, and Jeff Z. Pan. "Instances and Labels: Hierarchy-aware Joint Supervised Contrastive Learning for Hierarchical Multi-Label Text Classification." In The 2023 Conference on Empirical Methods in Natural Language Processing.

[2] Kumar, Ashish, and Durga Toshinwal. "HLC: hierarchically-aware label correlation for hierarchical text classification." Applied Intelligence 54, no. 2 (2024): 1602-1618.

**Questions:**

1. How is the proposed method differs from closely related existing methods?  Is it possible to compare with these methods quantitatively in the experiments?

2. Is the design of the proposed method medical-specific? Why do the authors write it as medical-specific but not a general hierarchical contrastive learning method?

3. Are there qualitative visualizations illustrate that hierarchy is preserved in the learned embedding space?

---

### Official Review · Reviewer_W2Ax · 2025-11-01

**Soundness:** 2
**Presentation:** 3
**Contribution:** 2
**Rating:** 2
**Confidence:** 4

**Summary:**

This paper addresses the significant disconnect between the flat-label assumption of standard contrastive learning and the natural hierarchical structure of labels in domains like medical imaging. The authors propose a framework with two plug-in objectives, Hierarchy-Weighted Contrastive and Level-Aware Margin, to inject this taxonomic information directly into the representation learning process. HWC is an elegant modification of supervised contrastive learning that scales logits within the softmax based on the normalized depth of the lowest common ancestor, thereby encouraging within-parent cohesion. LAM is a prototype-based margin loss that enforces separation between ancestor groups at different levels of the tree. The authors demonstrate experimentally on several medical and non-medical benchmarks that their combined approach outperforms current baselines (SimCLR, SupCon, HXE).

**Strengths:**

1. The framework is shown to be effective as a "drop-in" for standard Euclidean pipelines and also benefits from hyperbolic geometry, especially on deeper trees. This makes the method broadly applicable.
2. The ablation study in Figure 2 and Section 4.6 effectively disentangles the HWC gain from a simple temperature-tuning effect, supporting the authors' claim about the importance of pair-specific, in-softmax reshaping

**Weaknesses:**

1. The abstract mentions "self-supervised learning (SSL)", but the proposed HWC and LAM methods are fully supervised, requiring fine-grained leaf labels and the full label tree during pretraining. This should be made clearer from the outset. While the paper states it's "compatible with SSL using view-positives only", this claim is not experimentally verified.
2. The method introduces a non-trivial number of new hyperparameters. While typical ranges are provided, a comprehensive sensitivity analysis is missing. It's unclear how difficult it is to tune these parameters for a new dataset or taxonomy.
3. The paper presents its framework as 'geometry-agnostic' but heavily evaluates a hyperbolic variant, claiming it as a suitable setting for hierarchical data. However, the combination of hyperbolic geometry and contrastive learning, particularly for hierarchical data, is not a new concept. A significant body of prior work has already explored this exact intersection, including "Hyperbolic Contrastive Learning" (Ganea et al., 2023; arXiv:2302.01409), "Hyperbolic Hierarchical Contrastive Hashing" (Wang et al., 2022; arXiv:2212.08904), and, most notably, "Multi-Prototype Hyperbolic Learning Guided by Class Hierarchy" (Zhu et al., 2025; Springer:10.1007/s11263-025-02571-8). This last paper, in particular, appears to share significant conceptual overlap with the proposed LAM objective by using prototypes within a class hierarchy, also in hyperbolic space. The manuscript fails to adequately cite or differentiate itself from these highly relevant works. Without a clear articulation of its novel methodological contributions over these existing methods, the paper's primary contribution appears to be the application of these established ideas to medical imaging datasets. This positions the work more as a domain-specific application rather than a fundamental contribution to representation learning, which may be a mismatch for a general-purpose venue like ICLR.

**Questions:**

Especially weakness 3

---

### Note · Authors · 2025-11-14

I have read and agree with the venue's withdrawal policy on behalf of myself and my co-authors.